# Application of High-Performance Liquid Chromatography with Diode Array Detection to Simultaneous Analysis of Reference Antioxidants and 1,1-Diphenyl-2-picrylhydrazyl (DPPH) in Free Radical Scavenging Test

**DOI:** 10.3390/ijerph19148288

**Published:** 2022-07-07

**Authors:** Małgorzata Tatarczak-Michalewska, Jolanta Flieger

**Affiliations:** Department of Analytical Chemistry, Medical University of Lublin, Chodźki 4A, 20-093 Lublin, Poland

**Keywords:** HPLC, antioxidant activity, 1,1-diphenyl-2-picrylhydrazyl free radical, quercetin, resveratrol, Trolox, chlorogenic acid, hesperetin, coumarin

## Abstract

Antioxidant activity can be analyzed by various methods, both in vitro and in vivo. The widely used colorimetric method using the 1,1-diphenyl-2-picrylhydrazyl (DPPH) radical scavenging reaction has many limitations, such as interference from photosynthetic pigments naturally found in plant extracts. The DPPH-HPLC eliminates these troubles by enabling the separation of the DPPH free radical (DPPH-R) peak and its reduced form (DPPH-H) from other extract components. However, simultaneous analysis of antioxidants and evaluation of their activity is more complicated. To date, a post-column reaction with DPPH has been used for this purpose. The aim of the current study was the elaboration on a DPPH-RP-HPLC in gradient elution mode for simultaneous evaluation of the antioxidant activity of standards on the basis of DPPH-R peak inhibition, together with the identification of standards, as well as the products of redox reactions. The following antioxidants (AOs) were used as standards: quercetin, resveratrol, Trolox, chlorogenic acid, hesperetin, and coumarin. Flavone was used as the control chemical without antioxidant activity. The separation of the DPPH-R/DPPH-H pair, together with standards and reaction products, was studied on a C18 column using a gradient of acetonitrile from 5 to 60% within 20 min. The stability of DPPH was evaluated with different gradient profiles. The influence of the addition of acetic acid in concentrations of 0.05 to 1%, the duration of the analysis, and the radiation emitted by the UV lamp of a diode array detector on the induction of DPPH decomposition processes were investigated. The most significant parameter affecting DPPH stability appeared to be the acidic environment and water content in the mobile phase. An increase in the water content from 70 to 95% worsened the LOD of DPPH-R from 31.64 nM to 107.31 nM, as measured at 517 nm, and from 189.41 to 1677.05 nM at 330 nm. Each gradient profile provided good linearity (R^2^ = 0.9790–0.9977) of the relationship between the DPPH-R as well as DPPH-H peak areas, and a wide concentration range from 0.5 to 2.5 mM for UV-vis detection. Free radical scavenging activity was expressed by the percentage of DPPH-R peak inhibition at 517 nm. This simple method is suitable for monitoring DPPH radical scavenging by AO standards.

## 1. Introduction

Free radicals, which arise in the oxidation processes that take place both in food products and living organisms, lead to degradation, development of diseases, and acceleration of aging processes [1,2,3,4]. Therefore, free radical scavengers have become irreplaceable as additives in the food industry for food preservation, in pharmacology as stabilizers of medicinal preparations, and as dietary supplements supporting health. There are only a few synthetic antioxidants, such as butylated hydroxyanisole (BHA), butylated hydroxytoluene (BHT), and n-propyl gallate (PG). Plants and micro-organisms, including actinomycetes, bacteria, cyanobacteria, fungi, and lichens, are the main natural sources of exogenic antioxidants [5,6,7]. Antioxidant activity was confirmed for polyphenolic components (phenolic acids and flavonoids), carotenoids, vitamins (C, D, and tocopherols), and trace elements (Se, Zn, Cu, Mn, and Fe), as well as high-molecular-weight plant metabolites, such as tannins. Various spectroscopic, biochemical, and electrochemical assays are used to evaluate antioxidant capacity. There are many review articles in the literature that present the advantages and limitations of the methods used [8,9,10,11,12,13]. Many in vitro radical scavenging assays are based on the application of stable free radicals, such as 2,2-diphenyl-1-picrylhydrazyl radical (DPPH^•^), or free radicals generated in situ in the case of 2,2′-azino-bis(3-ethylbenzothiazoline-6-sulfonic acid (ABTS^•+^) and N,N-dimethyl-p-phenylene-diamine (DMPD^•+^), as well as nitric oxide (NO), superoxide (O2^•−^), and peroxynitrite (ONOO^−^) radicals. These assays use absorption of chromogenic agents at the visible length of light or fluorescence in the case of hydroxyl radical-averting capacity assay (HORAC).

The most popular antioxidant tests are related to the bleaching of DPPH, galvinoxyl, or the ABTS radical cation, as measured by electron spin resonance (ESR) both ex vivo and in vivo, as well as UV-vis spectrophotometry. Although the ESR method is suitable for detecting free radicals, it requires a complicated procedure [14,15]. In turn, fast and simple spectrophotometric measurement, as introduced by Blois [16], is subject to many other limitations. The colorimetric DPPH method relies on DPPH free radical absorption at a wavelength of 517 nm. After reaction with antioxidants, a drop in absorbance is observed due to 2,2-diphenyl-1-picrylhydrazine formation accompanied by a color change from purple to yellow. It is clear that in the case of complex samples, it is not possible to indicate which sample ingredients are responsible for the antioxidant effect. Moreover, compounds absorbing in the same wavelength range as DPPH can be a source of significant bias. A spectacular example is anthocyanins exhibiting strong absorption within the range of 500 to 550 nm, resulting in overlapping of their spectra with DPPH absorbing in the same wavelength range. The colorimetric method is also not intended to be used for analysis of colored foods due to interference caused by the added pigments [17]. JuDong Yeo and Fereidoon Shahidi [18] described the influence of pigments and colors in the extracts of plant-based foods on DPPH test results. The authors showed that spectral interference from coexisting pigments was at the level of 16.1% for colored extracts prepared from blackberry, raspberry, bell pepper, and beet.

The solution to the problem of spectral interference in colorimetric measurements is the use of paramagnetic electron resonance spectrometry (EPR) [18]. Another option is the DPPH radical scavenging assay coupled with high-performance liquid chromatography (HPLC) [17,18,19,20,21,22,23].

HPLC offers the possibility of detection of DPPH radicals and measurement of the area of its well-separated peak, thereby avoiding interference from color pigments of natural extracts. These attempts to use HPLC to test antioxidant activity with the DPPH scavenging test are based on the detection of the free radical DPPH or the free radical/reduced radical DPPH pair. This procedure requires the sample to be dosed twice, i.e., before and after the reaction of DPPH with the antioxidant. Chandrasekar et al. [21] developed an HPLC method for evaluating the DPPH free radical scavenging activity of commercial polyherbal formulations. The authors achieved DPPH free radical retention using a LiChrospher^®^ 100 RP-18e column (250 mm × 4 mm) and a mixture of methanol and water (80:20, *v*/*v*) as the mobile phase. The method was set up using various antioxidant standards (ascorbic acid, Trolox, probucol, and alpha-tocopherol); the downside is that the DPPH was determined only as one peak and not separately as radical and reduced forms. Yamaguchi et al. [17] reported measurement by HPLC using a TSKgel-Octyl-80Ts column with methanol–water (70:30, *v*/*v*) as mobile phase and detection at 517 nm. In 2012, Boudier et al. [22] obtained a similar result on a C18 column with a mobile phase containing acetonitrile-10 mM ammonium citrate buffer (pH 6.8; 70:30, *v*/*v*) and detection at 330 nm. Flieger et al. [23] developed an HPLC procedure for the simultaneous measurement of the DPPH redox pair (2,2-diphenyl-1-picrylhydrazyl radical)/DPPH-H (2,2-diphenyl-1-picrylhydrazine). Both forms were completely separated (Rs = 2.30, α = 1.65) on a Zorbax Eclipse XDB-C18 column eluted with methanol–water (80:20, *v*/*v*). The elaborated method was tested on standard antioxidants (AOs), i.e., reduced glutathione (GSH), ascorbic acid (vitamin C), and alcoholic extracts of *Aegopodium podagraria* L.

The approach associated with the identification of antioxidant components of complex mixtures and extracts by DPPH-HPLC requires more sophisticated instruments. For example, in 2007, Wu et al. [24] developed HPLC-ESI-MS and NMR to evaluate the antioxidant capacity of polyphenolic acids in plant extract. In turn, Nuengchamnong et al. [25] proposed RP-HPLC in conjunction with an MS/MS electrospray ionization system to identify antioxidant compounds in an extract of a Thai medicinal plant. In 2008, Tang et al. [26] described a rapid and simple method for screening natural antioxidants from the Chinese herb Flos Lonicerae Japonicae. The authors performed additional identification of antioxidants with the HPLC-DAD-TOF/MS hyphenated technique. The idea of the above approaches assumes that the peak areas of compounds with antioxidant activity are reduced after reaction with DPPH.

Another trend is the online detection of radical scavengers in HPLC eluates by a steady post-column photochemical reaction [27,28]. It was reported that the quenching of the chemiluminescence (CL) of a luminol signal enabled detection of the radical scavenging activity of antioxidants at the nanogram level. In turn, Koleva et al. [29] proposed online HPLC couplings with post-column derivatization with DPPH solution with the aim of evaluating the antioxidant activity of pure, natural antioxidants and aqueous extracts from Sideritis plants. The authors declared that the method was suitable for isocratic and gradient HPLC elution using a mobile phase modified with the addition of acetonitrile and methanol in concentrations of 10 to 90% and enriched with a buffer with a pH > 3. However, the use of post-column derivatization did not take into account the kinetics of the reaction between antioxidants and free radicals, which can be fast (<30 min) or slow (>1 h) [30]. Moreover, the separation of the DPPH radical from the reduced form was practically not achieved.

In our previous work [23], we used an isocratic elution with a low-water-content mobile phase. We measured the absorbance increases of the DPPH reduced form, as well as the DPPH free radical peak inhibition at UV-vis wavelengths. We observed that the visible light range was the most beneficial in terms of the sensitivity of the method and the range of linearity. The DPPH redox pair was stable under the applied conditions and underwent complete resolution. Unfortunately, the antioxidant standards, as well as extract components, were eluted at the beginning of the mobile phase due to high polarity. The aim of the present work was to assess the antioxidant activity of DPPH free radical scavenging with simultaneous diode array detection of the DPPH-R/DPPH-H redox pair, as well as standards (quercetin, resveratrol, Trolox, chlorogenic acid, hesperetin, and coumarin), and their reaction products with DPPH. Flavone was used as a control substance unable to react with DPPH. The research was preceded by analysis of the DPPH free radical stability under the conditions of analysis (gradient elution mode, acid addition to the mobile phase, UV radiation, analysis time, concentration of DPPH free radicals, and the concentration of antioxidant standards). The performed experiments updated knowledge about the influence of changing water content in ACN/water eluent systems on DPPH stability. The chromatograms were monitored at a wavelength of 517 nm, which is characteristic of a free radical form of DPPH, as well as at 330 nm, at which both forms exhibit significant absorbance.

## 2. Materials and Methods

### 2.1. Chemicals and Reagents

Acetonitrile of HPLC reagent grade and methanol of HPLC reagent grade were obtained from Merck (Darmstadt, Germany). Acetic acid 99.5% was obtained from POCH (Gliwice, Poland). 1,1-Diphenyl-2-picrylhydrazyl free radical (DPPH-R), quercetin, resveratrol, Trolox, chlorogenic acid, hesperetin, coumarin, and flavone were purchased from Sigma-Aldrich (St. Louis, MO, USA). Water purified by an ULTRAPURE Millipore Direct-Q 3UV-R (Merck, Darmstadt, Germany) with a resistivity of 18.2 MΩ cm was used to prepare all the aqueous solutions.

### 2.2. HPLC-DAD Conditions

HPLC analysis was carried out by Elite LaChrom HPLC Merck-Hitachi (Merck, Darmstadt, Germany) equipped with a detector array (L-2455), column thermostat Jetstream 2 Plus (Knauer, Berlin, Germany). Chromatographic separation was carried out at 20 °C using a ZORBAX Eclipse XDB-C18, 4.6 × 150 mm, 5 μm column (Agilent Technologies, Munich, Germany). Elution was carried out in linear gradient mode followed by an isocratic step. The mobile phases were composed of water (A) and acetonitrile (B). An increase in % B from 5%, 10%, 20%, and 30% B to 60% B was applied within 10 min; then, 60%B was continued for a subsequent 10 min. The addition of 1% acetic acid (CH_3_COOH) to A and B ensured a stable pH of 3.5 of the mobile phase in the case of investigation of the acidic environment with respect to DPPH stability. The flow rate of the mobile phase was 1 mL min^−1^. DAD detection enables chromatogram monitoring in the range of 200 to 800 nm. The injection volume was 20 µL, corresponding to the volume of the Rheodyne injector loop. The column void volume (dead time) was determined to be 1.34 min by injection of an unretained sample of uracil.

### 2.3. DPPH Free Radical Scavenging Activity Assay

A stock solution of 3 mM DPPH-R was prepared in methanol and stored in darkness at 4 °C in glass flasks. The working solutions were prepared daily by diluting the stock solution with methanol. Methanolic solutions (3 mM) of quercetin, resveratrol, chlorogenic acid, hesperetin, and coumarin were AOs. A 3 mM methanolic solution of flavone, which is the simplest member of the class of flavones consisting of 4H-chromen-4-one bearing a phenyl substituent at position 2, was used in the control test. The DPPH radical-scavenging activity was determined by mixing 417 µL of DPPH stock solution with 83 µL of the investigated standards. The above conditions ensured a constant DPPH concentration at a level of 2.5 mM. Measurements were performed after 50 min storage in the dark at room temperature (20 °C). HPLC-DAD chromatograms were obtained by gradient elution starting with 5, 10, 20, and 30% mobile phase B and ending with 60% mobile phase B within 20 min. The radical-scavenging activity of the DPPH-R by the antioxidant samples was evaluated using the following equation:The peak inhibition (%) = {(AC (0) − AA (t))/AC (0)} × 100(1)
where

AC (0) = peak area before the antioxidant addition; and

AA (t) = peak area after antioxidant addition at time.

### 2.4. Statistical Analysis

All statistical analyses were performed in the R statistical environment using R Core Team 2022 (Foundation for Statistical Computing, Vienna, Austria). The normality across groups was determined by the Shapiro–Wilk test. For each compound, the Welch ANOVA (analysis of variance) test was conducted to test for significant differences (*p* < 0.05) in the percentage of the DPPH peak inhibition with different gradient profiles. Additionally, Games–Howell post hoc tests were conducted, and the mean values of the peak inhibition for homogenic groups were established.

## 3. Results and Discussion

According to Blois [16], DPPH-R is stable only with respect to pH in the range of 5 to 6.5. Koleva et al. [29] expanded the factors influencing DPPH stability by including organic solvents such as methanol, acetonitrile, the addition of water, and buffers in the pH range of 2.2 to 6.6. Although the addition of water or buffer to the organic solvent caused decreased DPPH-R absorbance at 517 nm, HPLC in gradient elution mode with a mobile phase enriched with organic solvent from 10 to 90% was recommended. Moreover, the authors established that acidic buffers are useful for the improvement of HPLC resolution. It should be emphasized that the research carried out by Blois [16] concerned colorimetric experiments. In turn, Koleva’s experiments utilized post-chromatographic derivatization with a DPPH free radical. In preliminary experiments, we verified the influence of various factors on DPPH stability, such as water content in the mobile phase, acid addition to the mobile phase, and the initial concentration of the DPPH free radical.

### 3.1. Stability of DPPH-R in Gradient Elution Mode with Acetonitrile–Water Mobile Phase

A 1 mM methanolic solution of DPPH was analyzed in isocratic elution mode with an ACN/water (60:40, *v*/*v*) mobile phase, as well as with different gradient profiles, starting with 5% ACN/Water and ending with 60%ACN/water within 20 min. As shown in Figure 1, the reduced form of DPPH is always eluted faster in comparison to the free radical form. Both forms of DPPH can be identified by spectra recorded in the range of 200 to 800 nm, owing to the diode array detector (DAD). At 330 nm, both forms exhibit absorbance bands, whereas the band at 517 nm is characteristic only of free radicals. Therefore, to show peaks of the DPPH-R/DPPH-H pair, we chose 330 nm to monitor the chromatograms.

The ACN concentration applied at the beginning of the gradient influences the retention time of DPPH. Moreover, the increase in water content caused significant changes in the DPPH-R/DPPH-H peak area ratio measured at 330 nm. This is in agreement with a previous report, proving that water can promote free radical recombination [31]. The gradient profile starting with 5%ACN was unfavorable due to the completely disappearing free radical form, implying that high water content contributes to a reduction of 20 µL of 1 mM of DPPH-R. Figure 2 shows a comparison of the individual peak areas of DPPH-R and DPPH-H measured at 330 and 517 nm.

The measurements performed at 330 nm (Figure 2a) make it possible to observe the changes caused by the decrease in the water content in the mobile phase, both for the reduced form (DPPH-H) and for the free radical (DPPH-R). The resulting bars are inversely related to each other, which is the result of a balance between the components of the DPPH-R/DPPH-H pair. Thus, free radical quenching is related to an increase in the reduced form. On the other hand, at 517 nm (Figure 2b), the peak of the reduced form shows a constant and trace absorbance, which is obvious, considering that the absorption maximum of this substance at 517 nm approaches zero. Regarding the free radical, its peak regularly decreases as the water content of the eluent increases.

The changes observed in the peak areas for the DPPH-R/DPPH-H pair in different elution modes are reflected in the values of the detection limits (Table 1). The increase in water content is related to the deterioration of the detection limits, which reach higher values. Changing the water content from 95% to 70% at the beginning of the gradient worsens the LOD from three to ten times, depending on the chosen detection wavelength.

### 3.2. Influence of Analysis Time and UV Radiation Emitted by a Detector on DPPH Stability

The gradient elution profile experiments (Figure 1) undoubtedly showed the influence of the water content in the mobile phase on the ratio of DPPH forms and therefore the stability of the free radical. An alternative explanation for this could be the effect of the retention of the analyte and therefore its residence time on the column. There is a reasonable suspicion that the disappearance of the DPPH-R peak in the 5% acetonitrile elution as the initial composition of the mobile phase may be due to the much longer elution time. To verify the importance of the water content and the effect of analysis time, additional experiments were performed involving a variable composition of the mobile phase, ensuring a constant retention time of DPPH, which was achieved by regulating the flow rate.

By ensuring the constant retention of DPPH forms, we can relate the change in the DPPH forms ratio and thus DPPH-R stability only with water content in the mobile phase. The comparison presented in Figure 3 shows that the analysis time has a slight effect on DPPH decomposition; however, it appears to be more significant with the higher water content.

Another important factor causing the disappearance of the DPPH-R peak could be the photodegradation of free radicals. Scanning chromatograms in the range of 200–800 nm causes the high-energy UV radiation to pass through the detector cell, promoting photo-induced side processes in the sample. To evaluate this possible effect, we repeated experiments, setting detection without UV lamp ignition. As shown in Figure 3b, a diode array detector working at a wide range of wavelengths did not change the DPPH-R peak area in any of the gradient profiles studied.

### 3.3. Influence of DPPH-R Concentration on the DPPH-R/DPPH-H Ratio

Figure 4 shows the concentration-dependent curves for generation of the DPPH/DPPH-H ratio for 20 min with different gradient elution profiles starting with 5% ACN, 10% ACN, 20% ACN, or 30% ACN. Changes in the DPPH-R/DPPH-H ratio were measured at wavelengths characteristic of each form. The ratio increased with the increasing initial concentration of DPPH. The increasing response was obtained with up to 2 mM of DPPH. The DPPH-R/DPPH-H ratio maintained a constant level above 2 mM DPPH concentration, implying that examination of the antioxidant activity of any sample requires the addition of DPPH at a concentration exceeding 2 mM; only then can we be sure that the chromatographic process will not disturb the reducing action of antioxidants. It should be emphasized that the above requirement is related to the applied gradient elution conditions. For other gradient profiles, we can expect other values for the cutoff DPPH concentration.

### 3.4. Stability of DPPH Free Radical under Acidic Conditions

Koleva et al. [29] recommended the pH of the mobile phase to be higher than 3 to maintain DPPH solution stability in the HPLC system working in gradient elution mode. In the case of natural extract containing phenolic acids, acidic pH is commonly used to improve the RP-HPLC resolution. To verify the stability of 1mM DPPH under the conditions considered in this study, a stability test was performed using gradient elution starting from 30% ACN containing acetic acid in the range of 0.05 to 1% within 20 min. The results are shown in Figure 5. Unfortunately, additional unknown peaks appeared. preventing separation of the DPPH-R/DPPH-H pair. We observed a stepwise recombination of DPPH during chromatographic analysis of DPPH under the studied acidic conditions, regardless of the amount of acid.

This result calls into question a previous study reported by Koleva et al. [29]. The reason for these inconsistencies may be the longer duration time required to end the chromatographic process in our study in comparison to the short reaction time tested by Koleva, during which there were no significant changes in absorbance.

The effect of acetic acid concentration on the ratio of both forms of 2.5 mM DPPH solution measured at their characteristic wavelengths was also analyzed. As shown in previous experiments, this concentration ensures a constant ratio of both DPPH forms under the applied analysis conditions. CH_3_COOH concentrations were tested in the range of 0.05–1.0% (*v*/*v*)–0.05; 0.1; 0.25; 0.5; 0.75; 1.0% (*v*/*v*). As shown in Figure 6, an increase in acetic acid concentration in the mobile phase causes a decrease in the DPPH-R/DPPH-H ratio. Moreover, the addition of acetic acid worsens the precision of the measurements, as shown by the increasing standard deviation bars of the DPPH redox pair ratio with increased concentration. Therefore, further experiments were performed using an acetonitrile–water mobile phase.

### 3.5. Antioxidant Properties of AO Standards Measured by DPPH-HPLC-DAD

Antioxidant capacity can be defined as the difference in the area of the DPPH-R peak before and after reaction with AOs or as an increase in the DPPH-H peak area. As reported in previous studies, attempts to evaluate the DPPH-H/DPPH-R pair in the ultraviolet range or to measure the growth of the reduced form instead of inhibiting DPPH free radicals have been unsuccessful. It turns out that the range of visible light is the most favorable in terms of the sensitivity of the method and the range of linearity [23]. Therefore, the antioxidant activity of the tested compounds was determined as the % inhibition of the DPPH-R peak at 517 nm (Table 2). The examined gradient profiles did not significantly affect the obtained reducing effect of antioxidants. Although increased addition of water to the organic solvent in the mobile phase systematically lowered the DPPH-R absorbance at 517 nm, the obtained percentage inhibition values were within the assay error range (1.92–4.83%).

Comparison of average peak inhibition was performed via ANOVA. An ANOVA test requires verification of the assumptions of normality of distributions across groups, as well as the assumption of homogeneity of variance [32]. However, hypothesis testing for normality of the distribution across groups is ineffective with such a small sample size (three observations each), so tests of normality of the residuals, which are equivalent to testing for normality across groups, were conducted using the Shapiro–Wilk test. The descriptive statistics (Table 3) and Figure 7 show heterogeneity of variances in peak inhibition, but the Levene test does not rule out the hypothesis of homogeneity of variances. Despite the above and because the classic ANOVA test is less robust against heterogeneity of variance than non-normality, a Welch ANOVA test was conducted [33].

A Welch ANOVA test was conducted for each tested antioxidant. Table 3 shows significant differences in peak inhibition for chlorogenic acid, hesperetin, quercetin, and Trolox (*p* < 0.05). For these compounds, Games–Howell post hoc tests were conducted. The Games–Howell test is recommended when variance is not homogeneous [34]. Table 4 shows the mean values for selected homogenic groups.

Considering the percentage of the peak inhibition values, the antioxidant activity of the studied compounds can be arranged in the following order: quercetin > resveratrol > Trolox > chlorogenic acid > hesperetin > coumarin. The above sequence corresponds to the results obtained by other authors. So far, we have shown: the absence of antioxidant activity of flavone [35]; strong antioxidant activity of quercetin, exceeding that Trolox and resveratrol [36,37]; and weaker activity of hesperetin compared to quercetin [35]. Typically, a DPPH spectrophotometric technique is used to measure the antioxidant activity.

Among the studied AOs, the most effective is quercetin (Figure 8a,b), a flavonoid with well-known radical-scavenging properties that is often used as a reference compound in antioxidant tests. Its reaction with DPPH is rapid and stoichiometric [38]. The second is resveratrol (3,4′,5-trihydroxy-*trans*-stilbene) (Figure 8c,d), a natural phytoalexin found in grapes and wine that shows antioxidant and antiproliferative activities related to the presence of a hydroxyl group in the 3,4′,5 positions, as well as the double bond in the stilbene skeleton [39].

The free radical scavenging efficiency of Trolox (H2Tx; 6-hydroxy-2,5,7,8-tetramethylchromane-2-carboxylic acid) towards a wide variety of free radicals is high and proceeds through both single-electron transfer (SET) and hydrogen transfer (HT) mechanisms. Therefore, it has been commonly used as a reference antioxidant for assays covering ORAC (oxygen radical absorbance capacity), TRAP (total radical-trapping antioxidant parameter), FRAP (ferric reducing antioxidant power), TEAC (Trolox-equivalent antioxidant capacity), or other ABTS (2,20-azinobis(3-ethylbenzthiazoline-6-sulfonic acid) or DPPH assays. Trolox possesses two p*K*a values: 3.89 and 11.92 [40]. This means that in neutral pH, the carboxylate monoanionic form (HTx-1) is the dominant form, whereas below pH = 3, the neutral (H2Tx) form dominates, and above pH = 12, dianionic (Tx-2) forms of Trolox exist. Trolox is able to scavenge free radicals through different mechanisms. Besides HT and SET, we cannot exclude other mechanisms, such as RAF and SPLET:

HT: HTx-1 + R − Tx-1 + HR.

RAF: HTx-1 + R − [HTx–R]-1

SET: HTx-1 + R − HTx + R-1

SPLET: (i) HTx-1 “ H+ + Tx-2 (ii) Tx-2 + R − Tx-1 + R-1

In the case of chlorogenic acid, the number and location of hydroxyl groups are responsible for the ability to neutralize radicals. In reactions with free radicals, -OH groups act as hydrogen donors, leading to the formation of a phenoxy radical. It has been proven that the presence of the -OH group in the *ortho* position is responsible for improving the ability to neutralize free radicals [41].

Hesperetin ((*S*)-2,3-dihydro-5,7-dihydroxy-2-(3-hydroxy-4-methoxyphenyl) -4-benzopyran) is a flavanone belonging to citrus flavonoids that exhibits various promising biological activities, including anticancer, anti-inflammatory, neuroprotective, and antioxidant properties [42]. Nevertheless, of all the flavonoids, hesperetin acts as a weak scavenger of free radicals. This flavanon, like other flavonoids, possess a carbonyl group in its chemical structure and can be condensed with amino groups to form corresponding Schiff bases, which may be distinguished as derivatives with good radical-scavenging activities [43].

Coumarins, compounds derived from benzopyran, are quite similar to well-known antioxidant flavonoids and have been studied extensively for their antioxidant properties. Many coumarins have been isolated and identified from natural sources, especially green plants, and many others have been synthesized. Their protective effect against oxidative damage depends on the hydrogen-donating capacity of a hydroxyl group in each molecule [19].

In a previous paper, Furusawa and colleagues [44] reported the antioxidant activity of a series of flavonoids, mentioning the structure–activity relationship (SAR) of hydroflavonoids. The authors reported that unsubstituted flavone and three hydroxylated flavones were inactive in a DPPH assay. This confirms the specificity/selectivity of the DPPH test under the chromatographic conditions used in relation to the antioxidant substances. Moreover, they showed that flavanone, with hydrogenated C-2 and C-3, is completely inactive. Their results indicate that the presence of a double bond at the C-2 and C-3 positions was essential for the activity. In our study, flavone was used as the control substance (Figure 8g,h).

As shown in Figure 8, the proposed chromatographic system offers the opportunity to simultaneously analyze the DPPH-R/DPPH-H pair, the reference antioxidants, and the products of reaction with DPPH. On the chromatograms, we observed a reduction in the peak area of antioxidants following reaction with DPPH-R, as well as the surface of the DPPH-R peak at 517 nm. Flavone, as a control substance, does not react with DPPH-R, which is consistent with previous reports. The flavone and DPPH-R peaks before and after the reaction overlap with each other. This is also evidence that the chromatographic system and, in particular, the gradient profile used, were balanced in terms of DPPH-R consumption.

### 3.6. AGREE—Analytical GREEnness Metric Approach

An approach to analytical research that considers the environmental, health, and safety aspects of analytical procedures is known as green analytical chemistry (GAC). It turns out that the pro-ecological nature of analytical methodologies can be assessed using special GAC indicators. To date, several methods have been developed to determine these aspects, i.e., the National Environmental Methods Index (NEMI), the Analytical Eco-Scale, multi-criteria decision analysis (MCDA), and the Green Procedures Index. In recent years, a new metric system known as the RGB additive color model was developed. This model takes into account not only greenness criteria (green color) but also analytical performance (red color) and productivity (blue color). The final result, which is a combination of colors, is made up of indicators for each category [45].

To assess the greenness criteria of our study, we used the tool proposed in the work of Francisco Pena-Pereira in 2020 [46], thanks to the freely available software, the GREEnness analytical calculator (https://mostwiedzy.pl/AGREE, accessed on 11 April 2020). The final result is presented in the form of a pictogram. The assessment takes into account the 12 principles of green analytical chemistry. In each case, the user assesses on a scale of 0–1, assigning weights to the individual criteria. The antioxidant activity of a single reference substance using the RP-HPLC-DPPH method was assessed using the AGREE program. Equal weights were assigned to 12 criteria (principles 1–12), which proves that they are all equally important.

The procedure consists of preparing the sample, mixing it with DPPH solution, and then performing high-performance chromatographic separation of analytes with UV-vis detection. The procedure does not require sample processing, which qualifies by the reduced step-count criterion (principle 1). A semi-micro sample size is needed (principle 2). The measurement takes place offline (principle 3), and the procedure consists of five distinct steps (principle 4). The sample preparation procedure is neither automated nor miniaturized (principle 5). There are no derivatization factors involved in the analysis (principle 6). Total waste is 60 mL (two runs and column equilibration), consisting of methanol used for sample preparation and acetonitrile needed for the HPLC mobile phase. The analytical waste also contains 20 µL of 3 mM DPPH methanol solutions (11.82 mg/10 mL) and 20 µL of 3 mM ascorbic acid (5.28 mg/10 mL) (principle 7). The three analytes undergo retention in one run, and the sample throughput is ~3 samples h^−1^ based on a DPPH retention time of up to 16 min (principle 8). HPLC is the most energy-consuming analytical technique (principle 9), and solvents (alcohols) can come from biological sources (principle 10). The procedure requires the use of no less than 30 mL of toxic solvents (principle 11). Moreover, both methanol and acetonitrile are considered to be highly flammable and volatile liquids (principle 12). The results of the analysis are shown in Figure 9.

The results obtained for UV-vis spectrophotometric measurements are presented for comparison. Although the final ecological assessment is ultimately the same, despite individual differences, it should be emphasized that the weak point of the HPLC method is its higher consumption of energy, as well as toxic and flammable liquids. On the other hand, the advantages of HPLC include a smaller sample size with the possibility of identification, separation, and simultaneous evaluation of the activity of individual components in multicomponent samples.

## 4. Conclusions

The advantages of the current method, DPPH-RP-HPLC-DAD with gradient elution, include the possibility of simultaneous evaluation of antioxidant activity, together with the identification of standards and their products of reaction with DPPH (DPPH-R/DPPH-H) at different wavelengths. We studied, for the first time, the influence of water content in the mobile phase and analysis time on DPPH decomposition. Furthermore, we verified the possibility of photodegradation of DPPH by the high-energy, low-wavelength part of the spectrum. The limitation of the method is the analysis of very polar analytes, which undergo poor retention in the RP system. In this case, a less polar column would have to be used; however, the behavior of the DPPH free radical in this system would have to be re-examined.

Based on performed experiments, we can make the following recommendations: (i) The chromatographic system components consume DPPH-R. The examined system achieved equilibration at a DPPH concentration of 2 mM. Examining the antioxidant activity of any sample, the DPPH concentration should exceed this cutoff value. (ii) Addition of acetic acid to the mobile phase should be avoided, according to the observed DPPH-R recombination, into unknown products. (iii) Although the addition of water to the organic solvent decreases DPPH-R absorbance at 517 nm, HPLC in gradient elution mode with a mobile phase enriched with organic solvent from 5 to 60% did not significantly affect the % inhibition of the DPPH-R peak at 517 nm. The relative errors of the DPPH-R peak inhibition with the examined gradient profiles were less than 5%.

## Figures and Tables

**Figure 1 ijerph-19-08288-f001:**
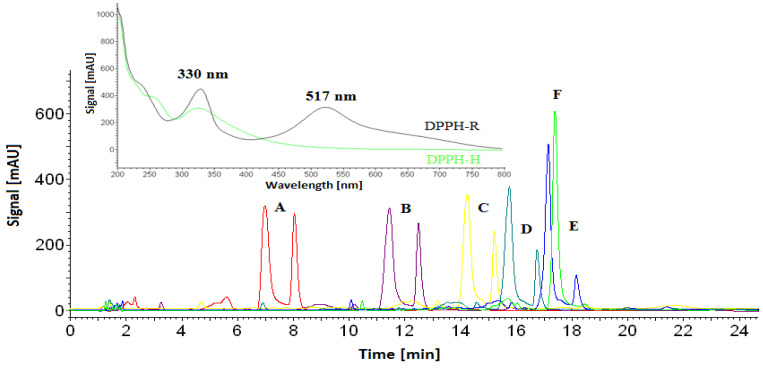
Overlaid chromatograms of 1 mM DPPH obtained by isocratic elution with 60% ACN/water (A, red) and gradient elution starting from 5%ACN (F, green), 20%ACN (E, blue), 30%ACN (D, light blue), 40% ACN (C, yellow), and 50% ACN (B, violet) and ending with 60% ACN within 20 min (detection, λ = 330 nm). Inserts show spectra of both forms recorded in the range of 200–800 nm by the DAD detector.

**Figure 2 ijerph-19-08288-f002:**
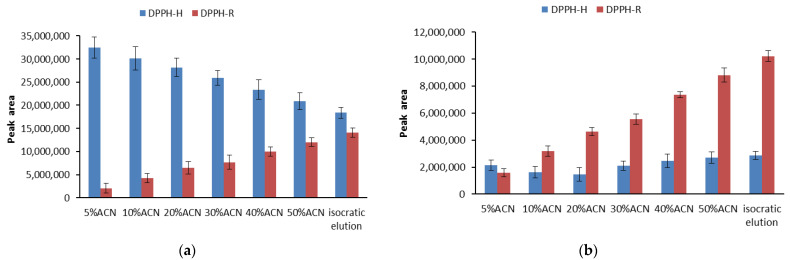
Comparison of the DPPH-R (red) and DPPH-H (blue) peak areas obtained using isocratic elution mode with 60%ACN, as well as gradient modes starting with 5%ACN, 10%ACN, 30%ACN, 40%ACN, and 50%ACN. The chromatograms were monitored at 330 nm (**a**) and 517 nm (**b**). The error bars indicating SD were calculated on the basis of three independent measurements.

**Figure 3 ijerph-19-08288-f003:**
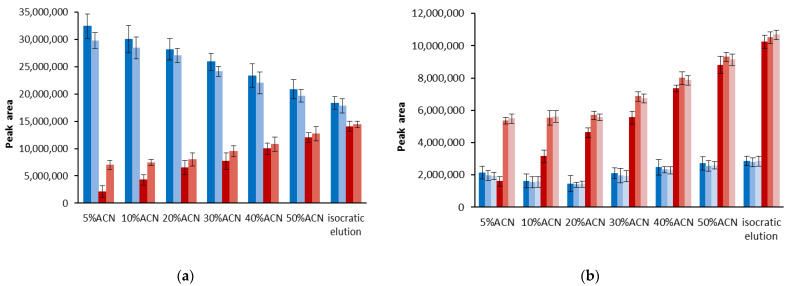
Comparison of the DPPH-R (brown) and DPPH-H (blue) peak areas obtained using isocratic elution mode with 60%ACN, as well as gradient modes starting with 5% ACN, 10% ACN, 30% ACN, 40% ACN, and 50% can, respectively. Dark-color columns represent gradient elution at a constant flow rate of 1 mL min^−1^, whereas light-color columns concern gradient profiles at a flow rate in the range of 0.8–2.3 mL min^−1^, ensuring a constant retention time of DPPH-H (11.38 ± 0.44 min) and DPPH-R (12.28 ± 0.53 min). The spectra were scanned in the range of 200–800 nm. The data were recorded at 330 nm (**a**) and 517 nm (**b**). The lightest columns in Figure 3b represent peak areas obtained by changing the flow rate of the mobile phase and recorded at 517 nm without UV lamp ignition. The error bars indicating SD were calculated on the basis of three independent measurements.

**Figure 4 ijerph-19-08288-f004:**
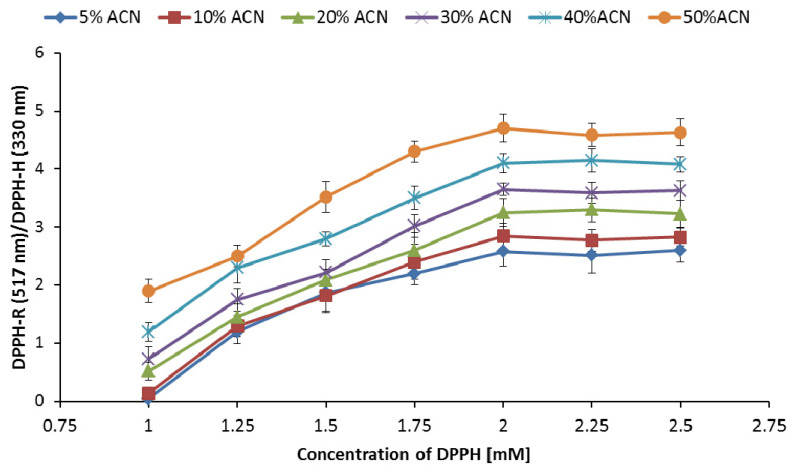
Effect of DPPH concentration on the DPPH-R/DPPH-H peak area ratio obtained using gradient modes starting with 5% ACN, 10% ACN, 20% ACN, 30% ACN, 40% ACN, and 50% ACN. The chromatograms were monitored at 330 nm (DPPH-H) and 517 nm (DPPH-R).

**Figure 5 ijerph-19-08288-f005:**
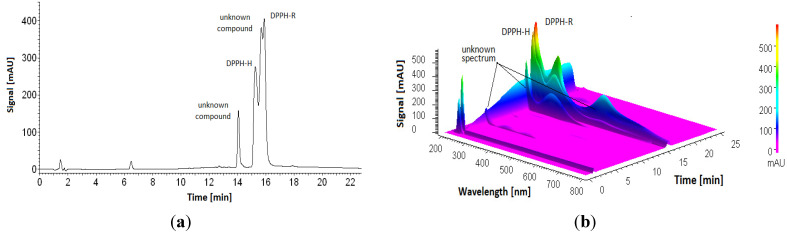
Chromatogram (2D (**a**) and 3D (**b**)) of 1 mM DPPH (**a**) obtained by gradient elution staring from 30% ACN/water with the addition of acetic acid within 20 min (pH = 3.5). The detection wavelength was set at 330 nm.

**Figure 6 ijerph-19-08288-f006:**
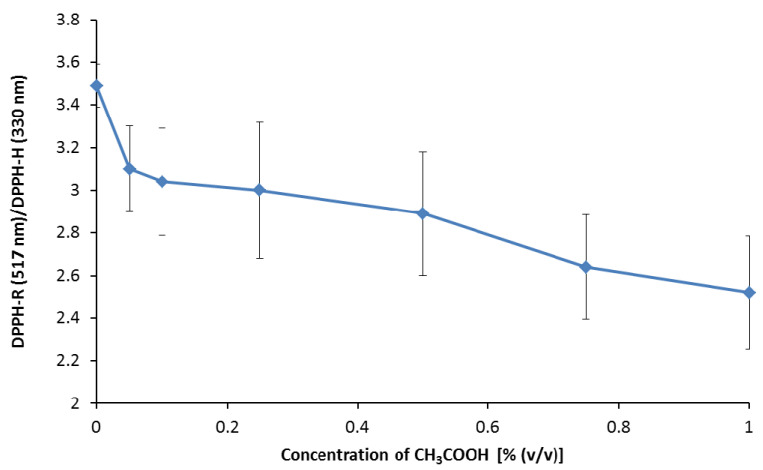
The influence of CH_3_COOH concentration on the DPPH-R/DPPH-H peak height ratio. Chromatographic conditions: gradient elution from 30% ACN to 60%ACN within 20 min and detection wavelengths of 330 nm (DPPH-H) and 517 nm (DPPH-R). The error bars indicating SD were calculated on the basis of three independent measurements.

**Figure 7 ijerph-19-08288-f007:**
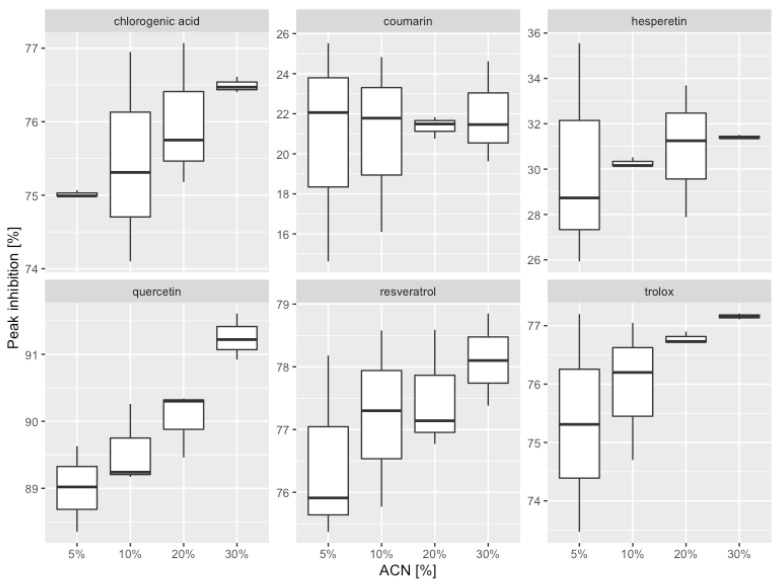
Box plots of peak inhibition within different gradient profiles.

**Figure 8 ijerph-19-08288-f008:**
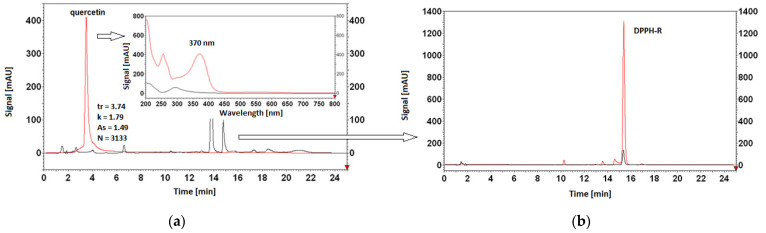
Overlaid spectra and chromatograms of methanolic solutions of standards: quercetin—λ = 370 nm (**a**), resveratrol—λ = 310 nm (**c**) hesperetin—λ = 290 nm (**e**), flavone—λ = 296 nm (**g**), coumarin—λ = 276 nm (**i**), chlorogenic acid—λ = 325 nm (**k**), Trolox—λ = 210 nm (**m**) before (red) and after reaction with DPPH (black), together with overlaid DPPH-R chromatograms—λ = 517 nm before (red) and after (black) reaction with the appropriate antioxidant: quercetin (**b**), resveratrol (**d**), hesperetin (**f**), flavone (**h**), coumarin (**j**), chlorogenic acid (**l**), and Trolox (**n**). Antioxidant standards were prepared by mixing 83 µL of a 3 mM solution of the given antioxidant and 417 µL of methanol. Conditions: an increase in acetonitrile from 30% to 60% within 10 min, then 60% ACN for a subsequent 10 min. The peaks of AOs are characterized by the retention time (t_r_), the retention parameter (*k*), tailing factor (A_s_), and the theoretical plate number values (N).

**Figure 9 ijerph-19-08288-f009:**
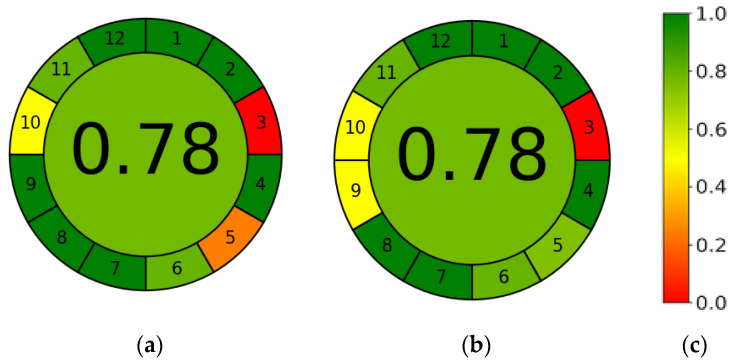
Results of AGREE analysis for UV-vis spectrometry (**a**) and HPLC-DAD (**b**). Each of the 12 principles of green chemistry is reflected with a color scale in the 0–1 range (**c**), while the weight of each principle is related to the width of its corresponding segment.The final assessment result is the color representation in the middle of a clock-like graph.

**Table 1 ijerph-19-08288-t001:** Parameters of the calibration curve investigated in the range of 0.5–2.5 mM, together with the limits of quantification (LOQ) and detection (LOD) for DPPH-R in gradient elution modes measured at 517 nm and 330 nm.

Gradient Profile	LOD [µmol L^−1^]	LOQ [µmol L^−1^]	Linearity Equation: (y = ax + b)				
			a ± SD	b ± SD	R^2^	F	s_e_
**λ = 517 nm**							
5% ACN	107.31	325.18	28,435,512 ± 935,113	−8,015,561 ± 1,444,263	0.9957	1,611,206	924.68
10% ACN	55.23	167.35	28,440,890 ± 1,303,644	−8,499,618 ± 2,013,452	0.9917	2,246,187	475.96
20% ACN	45.10	136.65	27,462,657 ± 1,417,624	−7,311,386 ± 2,189,491	0.9895	2,442,574	375.29
30% ACN	31.64	95.86	27,090,228 ± 1,681,039	−6,070,836 ± 2,596,330	0.9848	2,896,440	259.70
**λ = 330 nm**							
5% ACN	1677.05	5081.96	3,343,277 ± 81,109	−744,633 ± 125,272	0.9977	139,752	1699.04
10% ACN	642.58	1947.20	3,421,269 ± 132,552	−902,170 ± 204,725	0.9940	228,389	666.19
20% ACN	200.13	606.45	3,661,783 ± 245,725	−1,342,050 ± 379,518	0.9823	423,386	222.07
30% ACN	189.41	573.97	3,246,670 ± 237,833	−566,917 ± 367,329	0.9790	409,788	186.35

The REGLINP function was used to calculate the statistics for a straight line using the least-squares method; coefficient of determination (R^2^), Fisher F statistic (F), standard error of estimate (se), point of intersection (b ± SD), slope (a ± SD) with standard error values (SD) for constants. The LOD values (the detection limit) were calculated as LOD = 3.3 σ/S according to the ICH requirements, where σ is the standard deviation of the response measured as the standard error of the calibration curve, and S is the slope of the calibration curve. The limit of quantification (LOQ) was calculated as LOQ = 10 σ/S.

**Table 2 ijerph-19-08288-t002:** Free radical scavenging activity of the studied compounds. Sample preparation: 417 µL of 3 mM DPPH solution was mixed with 83 µL of 3 mM AO solution and incubated for 50 min in the dark at room temperature. Chromatographic conditions: gradient elution from 5, 10, 20, and 30% ACN to 60% ACN within 20 min and a detection wavelength of 517 nm.

ACN [% (*v*/*v*)]	AC (0) ^1^	AA (t) ^2^	Peak Inhibition		
			[%] ± SD	Mean [%]	Relative Error [%]
quercetin					
5%	56,744,541	6,236,679	89.01 ± 0.64	89.96	2.45
10%	56,920,403	5,940,782	89.56 ± 0.61		
20%	57,213,505	5,701,898	90.03 ± 0.50		
30%	58,620,394	5,129,109	91.25 ± 0.35		
resveratrol					
5%	56,550,363	13,284,246	76.51 ± 1.49	77.35	2.06
10%	56,725,622	12,896,570	77.27 ± 1.41		
20%	57,017,721	12,821,575	77.51 ± 0.96		
30%	58,419,796	12,782,251	78.12 ± 0.74		
Trolox					
5%	56,576,945	13,941,691	75.36 ± 1.87	76.33	2.33
10%	56,752,287	13,618,846	76.00 ± 1.19		
20%	57,044,523	13,246,879	76.78 ± 0.10		
30%	58,447,257	13,349,353	77.16 ± 0.05		
chlorogenic acid					
5%	56,608,329	14,141,327	75.02 ± 0.05	75.75	1.92
10%	56,783,768	13,921,676	75.48 ± 1.43		
20%	57,076,167	13,691,431	76.01 ± 0.97		
30%	58,479,679	13,748,572	76.49 ± 0.11		
hesperetin					
5%	56,768,577	39,634,117	30.18 ± 4.95	30.71	3.92
10%	56,944,512	39,710,256	30.27 ± 0.22		
20%	57,237,739	39,506,060	30.98 ± 2.92		
30%	58,645,224	40,224,759	31.41 ± 0.09		
coumarin					
5%	56,338,341	44,578,275	20.87 ± 5.57	21.29	4.83
10%	56,512,943	44,653,137	20.99 ± 4.43		
20%	56,803,947	44,675,168	21.35 ± 0.55		
30%	58,200,765	45,437,337	21.93 ± 2.52		

^1^ AC (0) = DPPH-R peak area before antioxidant addition (517 nm); ^2^ AA (t) = DPPH-R peak area after antioxidant addition (517 nm).

**Table 3 ijerph-19-08288-t003:** Welch ANOVA test results for each reference antioxidant.

Reference Antioxidant	*n*	Statistic	*p-*Value
chlorogenic acid	12	118.28	0.0004
coumarin	12	0.05	0.9830
hesperetin	12	17.27	0.0130
quercetin	12	10.13	0.0210
resveratrol	12	0.85	0.5300
Trolox	12	9.22	0.0330

**Table 4 ijerph-19-08288-t004:** Means of peak inhibition values obtained for the selected homogenic groups.

Compound	ACN	*n*	Mean	Homogenic Group	Mean Group 1	Mean Group 2
chlorogenic acid	5%	3	75.017	a	75.490	NA
chlorogenic acid	10%	3	75.453	ab	75.490	76.493
chlorogenic acid	20%	3	76.000	ab	75.490	76.493
chlorogenic acid	30%	3	76.493	b	NA	76.493
coumarin	5%	3	20.740	c	21.227	NA
coumarin	10%	3	20.903	c	21.227	NA
coumarin	20%	3	21.360	c	21.227	NA
coumarin	30%	3	21.903	c	21.227	NA
hesperetin	5%	3	30.073	de	30.427	31.413
hesperetin	10%	3	30.267	d	30.427	NA
hesperetin	20%	3	30.940	de	30.427	31.413
hesperetin	30%	3	31.413	e	NA	31.413
quercetin	5%	3	89.000	f	89.530	NA
quercetin	10%	3	89.557	fg	89.530	91.250
quercetin	20%	3	90.033	fg	89.530	91.250
quercetin	30%	3	91.250	g	NA	91.250
resveratrol	5%	3	76.487	h	77.329	NA
resveratrol	10%	3	77.217	h	77.329	NA
resveratrol	20%	3	77.500	h	77.329	NA
resveratrol	30%	3	78.110	h	77.329	NA
trolox	5%	3	75.327	ij	76.030	77.160
trolox	10%	3	75.983	ij	76.030	77.160
trolox	20%	3	76.780	i	76.030	NA
trolox	30%	3	77.160	j	NA	77.160

## Data Availability

Not applicable.

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
