# Peer review of "Application of High-Performance Liquid Chromatography with Diode Array Detection to Simultaneous Analysis of Reference Antioxidants and 1,1-Diphenyl-2-picrylhydrazyl (DPPH) in Free Radical Scavenging Test"

_ijerph, 2022, doi:10.3390/ijerph19148288_

Round 1
Reviewer 1 Report
The manuscript describes the development of a HPLC-DAD method to measure the radical scavenging power of antioxidants in a test based on DPPH. The manuscript is very well written and scientifically sound. The text as a whole is informative and authors comprehensively covered questions of DPPH stability in acidic pH and water proportion in the mobile phase on the detection of DPPH-R and DPPH-H before evaluating the selected antioxidants. My only question is why authors did not present the results of chlorogenic acid, coumarin and Trolox in Fig. 7? Therefore, I suggest acceptance once clearing this minor point.
Author Response
The manuscript describes the development of a HPLC-DAD method to measure the radical scavenging power of antioxidants in a test based on DPPH. The manuscript is very well written and scientifically sound. The text as a whole is informative and authors comprehensively covered questions of DPPH stability in acidic pH and water proportion in the mobile phase on the detection of DPPH-R and DPPH-H before evaluating the selected antioxidants. My only question is why authors did not present the results of chlorogenic acid, coumarin and Trolox in Fig. 7? Therefore, I suggest acceptance once clearing this minor point.
Thank you for this comment. We corrected figure 7 (now Fig.8) and added chromatograms of all studied antioxidants.
Reviewer 2 Report
The authors describe the separation of the DPPH-R/DPPH-H pair and antioxidant standards (quercetin, resveratrol, trolox, chlorogenic acid, hesperetin, coumarin, and flavone ) by HPLC with PDA detector on a C18 column. They tested the stability of DPPH-R in gradient elution (AcN/water), the influence of DPPH-R concentration, and its stability under acidic conditions. Finally, they measured the radical scavenging activity of the investigated compounds.
The article is well written but lacks originality. It is an extension of previously published work by the authors themselves (Flieger, Molecules 2020) or, for example, Koleva, Anal. Chem, 2000. Therefore, I do not recommend the article for publication.
I have one reservation about Figure 2b and their comments. It is clear that DPPH-H has a "constant and trace absorption" (l.240) when the absorption maximum of this substance at 517nm approaches zero.
The horizontal lines in the diagrams (Figs. 2, 3, 5) should be deleted.
Figure 4. unknown peaks: the authors do not specify whether unknown peaks occurred at all tested acetic acid concentrations or only at 1%.
Author Response
The authors describe the separation of the DPPH-R/DPPH-H pair and antioxidant standards (quercetin, resveratrol, trolox, chlorogenic acid, hesperetin, coumarin, and flavone ) by HPLC with PDA detector on a C18 column. They tested the stability of DPPH-R in gradient elution (AcN/water), the influence of DPPH-R concentration, and its stability under acidic conditions. Finally, they measured the radical scavenging activity of the investigated compounds.
The article is well written but lacks originality. It is an extension of previously published work by the authors themselves (Flieger, Molecules 2020) or, for example, Koleva, Anal. Chem, 2000. Therefore, I do not recommend the article for publication.
Thank You for this suggestion. The reviewer comment inspire us to perform additional experiments (suggested also by reviewer 3, and 4). We added some new aspects to our work considering the influence of analysis time on DPPH composition as well as its photodegradation. We added also subchapter considering the greenness aspects (suggested by rev.3). We do hope that added materials significantly improved the manuscript.
I have one reservation about Figure 2b and their comments. It is clear that DPPH-H has a "constant and trace absorption" (l.240) when the absorption maximum of this substance at 517nm approaches zero.
Yes We agree , so both spectrum as well our measurements are in agreement.
The horizontal lines in the diagrams (Figs. 2, 3, 5) should be deleted.-It was done.
Figure 4. unknown peaks: the authors do not specify whether unknown peaks occurred at all tested acetic acid concentrations or only at 1%.- The decomposition of DPPH appears in all cases. We gave just one of them as an example. It was explained in the text. Thank You for this observation. It was our mistake.

Reviewer 3 Report
The present manuscript entitled "Application of high-performance liquid chromatography with diode array detection to simultaneous analysis of reference antioxidants and 1,1-diphenyl-2-picrylhydrazyl (DPPH) in free radical scavenging test" by Małgorzata Tatarczak-Michalewska and Jolanta Flieger (ijerph-1779479) is written correctly and has a good structure; moreover, it has all the necessary parts. The article is interesting from an analytical and medical point of view; therefore, it should interest the reader. I proposed improvements in method description and with a presentation of figures. The paper meets the International Journal of Environmental Research and Public Health requirements. I recommend the article for publication in the International Journal of Environmental Research and Public Health following the common editing stage. My current decision is a minor revision. More specific comments and observations are presented below.
1. Abstract. Validation parameters can be added.
2. Introduction. The authors mentioned interferences. What can be done in the event of strong interference effects? How would you deal with them? What types of interference effects could occur?
3. Section 2.2. Was a precolumn used?
4. Figures with chromatogram and spectra. Axes should be well named with units, e.g., Time [min], Signal [mAU]. In the drawings with spectra, please, mark the bands that are mentioned in the text.
5. Figure 3. Please replace commas with dots. The name of the x-axis should also be added. Similarly, in other drawings.
6. Does the developed method have disadvantages? What are the limitations?
7. Conclusion. Please, emphasize clearly the advantages of the research carried out.
8. It would be worthwhile to evaluate the method using RGB Additive Color Model to Analytical Method Evaluation or AGREE-Analytical GREEnness Metric Approach.
I hope that the comments presented will help improve the article.
Author Response
The present manuscript entitled "Application of high-performance liquid chromatography with diode array detection to simultaneous analysis of reference antioxidants and 1,1-diphenyl-2-picrylhydrazyl (DPPH) in free radical scavenging test" by Małgorzata Tatarczak-Michalewska and Jolanta Flieger (ijerph-1779479) is written correctly and has a good structure; moreover, it has all the necessary parts. The article is interesting from an analytical and medical point of view; therefore, it should interest the reader. I proposed improvements in method description and with a presentation of figures. The paper meets the International Journal of Environmental Research and Public Health requirements. I recommend the article for publication in the International Journal of Environmental Research and Public Health following the common editing stage. My current decision is a minor revision. More specific comments and observations are presented below.
- Abstract. Validation parameters can be added.
We added additional sentences to abstract part.
- Introduction. The authors mentioned interferences. What can be done in the event of strong interference effects? How would you deal with them? What types of interference effects could occur?
Thank You for this suggestion. We added the following sentences and additional reference to answer this question:
JuDong Yeo and Fereidoon Shahidi [18] described the influence of pigments and colors in the extracts of plant-based foods on the DPPH test result. The authors showed that the spectral interference from coexisting pigments was on the level of 16.1% for colored ex-tracts prepared from Blackberry, Raspberry, Bell Pepper, and Beet.
The solution to the problem of spectral interference in colorimetric measurements is the use of paramagnetic electron resonance spectrometry (EPR) [18]. Another option is the DPPH radical scavenging assay coupled with high-performance liquid chromatography (HPLC) [17-23].
- Yeo, J.; Shahidi, F. Critical Re-Evaluation of DPPH assay: Presence of Pigments Affects the Results. Agric. Food Chem. 2019, 67, 7526-7529. doi: 10.1021/acs.jafc.9b02462.
- Section 2.2. Was a precolumn used?
No there was no precolumn in our experiments used, that is why we don’t mention it in the methodological section.
- Figures with chromatogram and spectra. Axes should be well named with units, e.g., Time [min], Signal [mAU].In the drawings with spectra, please, mark the bands that are mentioned in the text.
The figures were corrected accordingly to this suggestion.
- Figure 3. Please replace commas with dots.The name of the x-axis should also be added. Similarly, in other drawings.
The figures were corrected accordingly to this suggestion.
- Does the developed method have disadvantages? What are the limitations?
The limitation of the method is the analysis of very polar analytes, which undergo poor retention in the RP system. In this case, a less polar column would have to be used and the behavior of the DPPH free radical in this system would have to be re-examined.
- Conclusion. Please, emphasize clearly the advantages of the research carried out.
The advantages of the current method DPPH-RP-HPLC-DAD with gradient elution cover the possibility of the simultaneous evaluation of the antioxidant activity together with the identification of standards as well as their products of reaction with DPPH (DPPH-R/DPPH-H) at different wavelengths. We studied for the first time influence of water content in the mobile phase, and analysis time on DPPH decomposition. Furthermore, we checked the possibility of photodegradation of DPPH by the high-energy low wavelength part of the spectrum.
- It would be worthwhile to evaluate the method using RGB Additive Color Model to Analytical Method Evaluation or AGREE-Analytical GREEnness Metric Approach.
The new subchapter devoted to RGB Additive Color Model to Analytical Method Evaluation was added.:
3.4. AGREE-Analytical GREEnness Metric Approach
An approach to analytical research that considers the environmental, health, and safety aspects of analytical procedures is known as green analytical chemistry (GAC). It turns out that the pro-ecological nature of analytical methodologies can be assessed using special GAC indicators. So far, several methods have been developed to determine them, ie National Environmental Methods Index (NEMI), Analytical Eco-Scale, multi-criteria decision analysis (MCDA), Green Procedures Index, Analytical. In recent years, a new metric system known as the RGB additive color model has been developed. This model takes into account not only greenness criteria (green color), but also analytical performance (red color) and productivity (blue color). The final result, which is a combination of colors, is made up of indicators for each category [45].
To assess the greenness criteria of our study, we used the tool, proposed in the work of Francisco Pena-Pereira in 2020 [46], thanks to the freely available software for the GREEnness analytical calculator (https://mostwiedzy.pl/AGREE.). The final result is created in the form of a pictogram. The assessment takes into account the 12 principles of green analytical chemistry. In each case, the user assesses on a scale of 0-1, assigning weights to the individual criteria. Determination of the antioxidant activity of a single reference substance using the RP-HPLC-DPPH method was assessed using the AGREE program. Equal weights were assigned to 12 criteria (principles 1-12), which proves that they are all equally important.
The procedure consists in preparing the sample, mixing it with DPPH solution, and then performing high-performance chromatographic separation of analytes with UV-Vis detection. The procedure does not require sample processing which qualifies by the reduced step count criterion (principle 1) and a semi-micro sample size is needed (principle 2). The measurement takes place offline (principle 3) and the procedure consists of five distinct steps (principle 4). The sample preparation procedure is neither automated nor miniaturized (principle 5). There are no derivatization factors involved in the analysis (principle 6). Total waste is 60 ml (two injections with column equilibration), consisting of dosed samples, methanol used for sample preparation, 20 ml acetonitrile for the HPLC mobile phase. The analytical waste consists of 20microL ml of 3mM DPPH methanol solutions (11.82 mg / 10 mL) and 20microL ml o of 3 mM ascorbic acid (5.28 mg / 10 mL) (principle 7). The three analytes undergo retention in one run and the sample throughput is ~ 3 samples h – 1 based on a DPPH retention time of up to 16 min (principle 8). HPLC is the most energy-consuming analytical technique (principle 9) and solvents (alcohols) can come from biological sources (principle 10). The procedure requires the use of not less than 30 ml of toxic solvents (principle 11). Moreover, both methanol and acetonitrile are considered to be highly flammable and volatile liquids (principle 12). The results of the analysis are shown in Figure 8.
|
(a) |
(b) |
Figure 8. Results of AGREE analysis for UV-vis spectrometry (a) and HPLC-DAD (b).
For comparison, the results obtained by means of UV-vis spectrophotometric measurements are presented. Although the ecological assessment is ultimately the same, it should be emphasized that the weak point of the HPLC method is the greater consumption of energy, as well as toxic and flammable liquids. On the other hand, the advantages of HPLC include a smaller sample size with the possibility of identification, separation and simultaneous evaluation of the activity of individual components in multicomponent samples.
I hope that the comments presented will help improve the article.
Thank You once again for all comments and time devoted to our work.

Reviewer 4 Report
The manuscript by M. Tatarczak-Michalewska and J. Flieger is devoted to the solving the problem of adequate assessment of natural compounds antioxidant activity (DPPH test) in complex mixtures using HPLC-DAD technique. Although the work is not without some merit and the presented results are of some practical interest, the serious concerns on methodology and data interpretation have arisen after closer examination preventing me from recommending the publication of manuscript in its present form:
1. In experiments with gradient elution profile (Fig. 1) the authors associate the change in DPPH forms ratio and thus DPPH-R stability with water content in the mobile phase. However, an alternative explanation based on the change in elution time can be proposed. In other words, the disappearing the DPPH-R peak in elution with 5% acetonitrile as initial mobile phase composition can be caused by much longer analysis time enough for analyte decomposition. To discriminate water content and analysis time effects, additional experiments involving the varying mobile phase composition at constant retention time (due to different flowrates) are required.
2. Another important factor can be photodegradation of DPPH-R. The authors use diode array detector in which the entire radiation of UV lamp including high-energy low wavelength part of the spectrum passes through the sample promoting photo-induced side processes. To assess this possible effect, I would recommend performing detection without UV lamp ignition (only VIS halogen radiation source) or using scanning spectrophotometric detector with monochromatic light passing through the cell.
3. Conclusions on the DPPH forms ratios made from Fig. 2 seem unjustified. The ratio of extinction coefficients at 330 and 517 nm is constant for each compound, thus the substantial decrease of DPPH-H chromatographic peak area measured at 330 nm must be accompanied with the proportional decrease at 517 nm which is not observed in Fig. 2b (the peak area even increases there!). This can be explained with the formation of other co-eluting compounds contributing to the absorbance. An evidence for such possibility is the spectral measurements in the presence of acetic acid shown in Fig. 4. If the formation of other compounds is really possible, they must be taken into account in the interpretation of chromatographic data to make antioxidant activity measurements more reliable.
4. Since DPPH contains nitrogen atoms and can be easily protonated (DPPH-H has pKa of 8.6) with substantial changes in UV-VIS spectra and probably chromatographic retention the mobile phase must be buffered and must provide enough buffer capacity. However, as I understood from the experimental section, pH was not maintained in experiments except the study of acetic acid effect. The alternative interpretation of Fig. 5 can be based on the effect of pH on the DPPH protolytic equilibria and thus absorbance but not on the radical and H-forms ratios.
5. The retention of antioxidants achieved in this study and demonstrated on Fig. 7 is relatively low. The authors do not provide the retention factors or chromatographic system void volumes, however, one can assume that the separation of resveratrol and even quercetin from dead volume is not sufficiently good foe the analysis of real samples of complex natural objects. In this situation the use of stationary phases with mixed retention mechanism (for example, octadecyl with embedded polar amide groups) can be recommended.
6. The accuracy of the data representation (number of significant figures) in the tables is completely redundant.
Author Response
The manuscript by M. Tatarczak-Michalewska and J. Flieger is devoted to the solving the problem of adequate assessment of natural compounds antioxidant activity (DPPH test) in complex mixtures using HPLC-DAD technique. Although the work is not without some merit and the presented results are of some practical interest, the serious concerns on methodology and data interpretation have arisen after closer examination preventing me from recommending the publication of manuscript in its present form:
- In experiments with gradient elution profile (Fig. 1) the authors associate the change in DPPH forms ratio and thus DPPH-R stability with water content in the mobile phase. However, an alternative explanation based on the change in elution time can be proposed. In other words, the disappearing the DPPH-R peak in elution with 5% acetonitrile as initial mobile phase composition can be caused by much longer analysis time enough for analyte decomposition. To discriminate water content and analysis time effects, additional experiments involving the varying mobile phase composition at constant retention time (due to different flowrates) are required.
Thank you very much for this suggestion. We performed additional experiments involving the varying mobile phase composition at constant retention time of DPPH forms . We described the results in subchapter 3.2.
3.2. Influence of the analysis time, and the UV radiation emitted by a detector on the DPPH stability.
The gradient elution profile experiments (Fig. 1) undoubtedly showed the influence of the water content in the mobile phase on the ratio of DPPH forms, and thus the stability of the free radical. An alternative explanation for this could be the effect of the retention of the analyte and therefore its residence time on the column. There is a reasonable suspicion that the disappearance of the DPPH-R peak in the 5% acetonitrile elution as the initial composition of the mobile phase may be due to the much longer elution time. To verify the importance of the water content and the effect of the analysis time, additional experiments were performed involving a variable composition of the mobile phase ensuring a constant retention time of DPPH, which was achieved by regulating the flow rate.
|
(a) |
(b) |
Figure 3. Comparison of the DPPH-R (brown), and DPPH-H (blue) peaks’ area obtained using isocratic elution mode with 60%ACN as well as gradient modes starting with 5%ACN, 10%ACN, 30%ACN, 40%ACN, 50%ACN respectively. Dark color columns represent gradient elution at constant flow rate of 1 mL min-1, whereas light color columns concern gradient profiles at flow rate in the range of 0.8 - 2.3 mL min-1 ensuring constant retention time of DPPH-H (11.38 ± 0.44 min) and DPPH-R (12.28 ± 0.53 min). The spectra were scanned in the range of 200-800 nm. The data were recorded at 330 nm (a), and 517 nm (b). the lighest columns in Fig.3b represent peak areas obtained with changing the flow rate of the mobile phase, and recorded at 517 nm without UV lamp ignition. The error bars indicating SD were calculated on the basis of three independent measurements.
By ensuring the constant retention of DPPH forms we can relate the change in DPPH forms ratio and thus DPPH-R stability only with water content in the mobile phase. The comparison presented in Fig.3 shows that the analysis time has a slight effect on DPPH decomposition, however, it appears to be more significant at the higher water content.
Another important factor causing the disappearance of the DPPH-R peak can be the photodegradation of free radicals. Scanning chromatograms in the range of 200-800 nm causes the high-energy UV radiation to pass through the detector cell promoting photo-induced side processes in the sample. To evaluate this possible effect, we repeated experiments setting detection without UV lamp ignition. As can be seen in Fig.3b a diode array detector working in a wide range of wavelengths did not change the DPPH-R peak area at any of the gradient profiles studied.
- Another important factor can be photodegradation of DPPH-R. The authors use diode array detector in which the entire radiation of UV lamp including high-energy low wavelength part of the spectrum passes through the sample promoting photo-induced side processes. To assess this possible effect, I would recommend performing detection without UV lamp ignition (only VIS halogen radiation source) or using scanning spectrophotometric detector with monochromatic light passing through the cell.
Thank you very much for this suggestion. We performed additional experiments without UV lamp . We described the results in subchapter 3.2.
- Conclusions on the DPPH forms ratios made from Fig. 2 seem unjustified. The ratio of extinction coefficients at 330 and 517 nm is constant for each compound, thus the substantial decrease of DPPH-H chromatographic peak area measured at 330 nm must be accompanied with the proportional decrease at 517 nm which is not observed in Fig. 2b (the peak area even increases there!). This can be explained with the formation of other co-eluting compounds contributing to the absorbance. An evidence for such possibility is the spectral measurements in the presence of acetic acid shown in Fig. 4. If the formation of other compounds is really possible, they must be taken into account in the interpretation of chromatographic data to make antioxidant activity measurements more reliable.
We agree with the reviewer that at the first look it seems obscure, however, looking at the spectra inserted in Fig.1 we can observe that the reduced form of DPPH assigned as DPPH-H has an absorption maximum approaches zero at 517nm. In this way, we can explain rather constant and trace absorption of DPPH-H at 517 nm (Fig.2b). In each case, we checked the spectra purity, so it would be difficult to prove the contamination of peaks. Another situation concerns acidic pH, which is not disadvantageous because of the decomposition of free radicals as was shown on chromatograms.
The reviewer's suggestion is important and shows us that this part of the manuscript should be completed with additional explanation.
- Since DPPH contains nitrogen atoms and can be easily protonated (DPPH-H has pKa of 8.6) with substantial changes in UV-VIS spectra and probably chromatographic retention the mobile phase must be buffered and must provide enough buffer capacity. However, as I understood from the experimental section, pH was not maintained in experiments except the study of acetic acid effect. The alternative interpretation of Fig. 5 can be based on the effect of pH on the DPPH protolytic equilibria and thus absorbance but not on the radical and H-forms ratios.
We thank the reviewer for this valuable suggestion. Buffers, especially in the gradient elution, are used where they are important for the separation. In our case, they were not necessary. Besides, they would complicate the system by introducing a new variable in the form of changing ionic strength. In an aqueous-organic system, such as that found in a chromatographic system, one cannot assume only acid-base reactions, because the mechanism is mixed, i.e. HAT / SET. We only checked the addition of acid for the stability of DPPH, i.e. whether the ratio of DPPH-H / ​​DPPH-R forms changes, because this acid is often a component of the mobile phase in the analysis of samples containing polyphenols, flavonoids, and therefore popular antioxidants. In addition, a buffer with pH 6.6 may result in suspension when mixed with DPPH solutions, hence the concern about introducing the buffer in the mobile phase gradient. We agree with the reviewer that a buffer with a pH of about 7 can be introduced into the sample prior to loading onto the column when the antioxidant / DPPH-R reaction is taking place in order to simulate possible physiological conditions in the body, as is often used.
- The retention of antioxidants achieved in this study and demonstrated on Fig. 7 is relatively low. The authors do not provide the retention factors or chromatographic system void volumes, however, one can assume that the separation of resveratrol and even quercetin from dead volume is not sufficiently good foe the analysis of real samples of complex natural objects. In this situation the use of stationary phases with mixed retention mechanism (for example, octadecyl with embedded polar amide groups) can be recommended.
We added chromatographic parameters like tr, As, N, k, tr in Fig. 8 (Fig.7 from the previous version). Into the methodological part, we added information concerning the void volume. We agree with the reviewer that analytes with higher polarity could be eluted too quickly. So, for other analytes, another gradient should be optimized or a less hydrophobic stationary phase should be applied. Thank you for this comment. We added this future perspective for researchers interesting in the continuation of DPPH-HPLC studies.
- The accuracy of the data representation (number of significant figures) in the tables is completely redundant.
Thank you for this comment. We prepared new rounding of some numbers. However some of them we can not change because they express peak areas that is why they are so huge.

Round 2
Reviewer 2 Report
The manuscript was improved by additional experiments (addition of acetic acid from 0.05 to 1%, the duration of the analysis, and the radiation emitted
by the UV lamp of a diode array detector on the induction of DPPH decomposition process) and also analytical greenness metric approach was performed. These experiments slightly improved the novelty of the paper and I think, that it can be published. But there are still some typos, especially in paragraph 3.6
Author Response
Thank you very much for your approval. Frankly, we made some typos, especially in paragraph 3.6. They were corrected. The authors appreciate the reviewer's comments and suggestions and efforts to improve our manuscript.
Once again the authors would like to express our thanks.
Reviewer 4 Report
The authors followed the reviewer's comments and performed additional experiments to clarify the key issues. Now I can recommend the manuscript for publication.
Author Response
Thank you very much for your approval. The authors appreciate all reviewer's comments and suggestions and efforts to improve our manuscript.